# Stable single atomic silver wires assembling into a circuitry-connectable nanoarray

Yaxin Chen [1,11], Daiming Tang [2,11], Zhiwei Huang[1,3,11], Xi Liu[4,5,11], Jun Chen[2], Takashi Sekiguchi[2,6], Weiye Qu [1], Junxiao Chen[1], Dongrun Xu[1], Yoshio Bando [2,7,8], Xiaolei Hu[1], Xiaoping Wang[5], Dmitri Golberg[2,9] & Xingfu Tang [1,10 ✉]

Atomic metal wires have great promise for practical applications in devices due to their unique electronic properties. Unfortunately, such atomic wires are extremely unstable. Here we fabricate stable atomic silver wires (ASWs) with appreciably unoccupied states inside the parallel tunnels of $\alpha$-MnO$_2$ nanorods. These unoccupied Ag $4d$ orbitals strengthen the Ag–Ag bonds, greatly enhancing the stability of ASWs while the presence of delocalized $5s$ electrons makes the ASWs conducting. These stable ASWs form a coherently oriented three-dimensional wire array of over 10 nm in width and up to 1 μm in length allowing us to connect it to nano-electrodes. Current-voltage characteristics of ASWs show a temperature-dependent insulator-to-metal transition, suggesting that the atomic wires could be used as thermal electrical devices.

[1] Department of Environmental Science and Engineering, Fudan University, Shanghai, China. [2] World Premier International Center for Materials Nanoarchitectonics, National Institute for Materials Science, Namiki 1-1, Tsukuba, Ibaraki, Japan. [3] Department of Environmental Science and Engineering, College of Chemical Engineering, Huaqiao University, Xiamen, Fujian, China. [4] School of Chemistry and Chemical, In-situ Center for Physical Science, Shanghai Jiao Tong University, Shanghai, China. [5] Syncat@Beijing, SynfuelsChina Co. Ltd, Beijing, China. [6] Faculty of Pure and Applied Science, University of Tsukuba, Tsukuba, Japan. [7] Australian Institute for Innovative Materials, University of Wollongong, Wollongong, NSW, Australia. [8] Institute of Molecular Plus, Tianjin University, Tianjin, China. [9] Centre for Materials Science and School of Chemistry and Physics, Queensland University of Technology (QUT), Brisbane QLD, Australia. [10] Shanghai Institute of Pollution Control and Ecological Security, Shanghai, China. [11] These authors contributed equally: Yaxin Chen, Daiming Tang, Zhiwei Huang, Xi Liu. ✉ email: tangxf@fudan.edu.cn

One-dimensional (1D) atomic metal wires are promising materials as building blocks in electronic nanodevices because their electronic characteristics are distinct from those of their bulk counterparts[1–5]. For practical applications of atomically thin metal wires in nanoelectronics, high stability in air at room temperature and a suitable length for convenient connection into nanocircuitry are key requirements. However, the high degree of coordinative unsaturation makes the formation of long atomic metal wires extremely challenging.

Over the past two decades, the creation of long atomic metal wires through (self-)assembly techniques have been reported only in a few cases. Atomic silver wires (ASWs) of up to 100 nm in length were synthesized on Pt(997) surfaces under ultrahigh vacuum conditions[6]. Infinite linear atomic metal wires were fabricated in the tunnels of an air-sensitive inorganic subnitride[7]. Using a scanning tunneling microscope, 1D atomic wires of gold, silver, or manganese with controllable lengths were constructed on surfaces by self-assembling[3,5,8]. However, these long atomic wires are unstable, and thus difficult to use in practical applications[7–9]. Owing to the atomically thin wire width, assembling many long atomic wires as a coherently oriented array is highly desirable for constructing miniaturized device architectures.

Herein, we create stable ASWs of up to 1 µm in length by a self-assembly technique, leading to a coherently oriented three-dimensional (3D) array inside the tunnels of an insulating α-MnO₂ nanorod, the stability of which derives from the scaffolding function of α-MnO₂ and the strengthened Ag–Ag bonds due to the appreciably unoccupied states of the Ag 4d orbitals. The stable atomic wire array with suitable 3D sizes can be conveniently connected with nanoelectrodes for conductance measurements. Current-voltage (I–V) data demonstrate a temperature-controlled insulator-to-metal transition, making ASWs attractive for application as thermal electrical devices.

## Results

**Geometric structures of the atomic wire array**. ASWs were synthesized by a thermal diffusion method starting from Ag nanoparticles (NPs) supported on surfaces of the α-MnO₂ nanorods (see the "Methods" section for more details). The accurate geometrical structures of ASWs were investigated by synchrotron X-ray diffraction (SXRD) patterns, extended X-ray absorption fine structure (EXAFS) spectroscopy and transmission electron microscopy (TEM) imaging. TEM data shows that the incorporation of Ag atoms hardly changed the morphology of α-MnO₂ nanorods (Supplementary Figs. 1 and 2). We further conducted a Rietveld refinement of room-temperature SXRD of ASWs inside the α-MnO₂ tunnels together with the pristine α-MnO₂ (Supplementary Fig. 3)[10]. The resulting lattice information and structural parameters were summarized in Supplementary Tables 1 and 2. These data indicate that the Ag atoms of ASWs are located at the Wychoff 2a sites inside the α-MnO₂ tunnels[11], and thus the closest Ag–Ag and Ag–O distances are determined to be 2.87 and 2.48 Å, respectively.

The local structure of Ag atoms in ASWs was further explored by EXAFS spectra at the Ag K-edge using both Fourier and wavelet transforms (Fig. 1). The wavelet transform plots show that the maxima of the wavelet transforms of Ag foil and Ag₂O are ~8 and ~7 Å⁻¹, which correspond to Ag–Ag and Ag–O scattering paths, respectively. Compared with the two references, ASWs have k maxima at ~5, ~6, and ~7 Å⁻¹, which should correspond to the first-shell Ag–O, metallic Ag–Ag, and second-shell Ag–O/Mn scattering paths, respectively. To verify this inference, we conducted EXAFS fitting to unveil the quantitative coordination configuration of Ag atoms. The fitting results show that in ASWs the interatomic distances in the two nearest

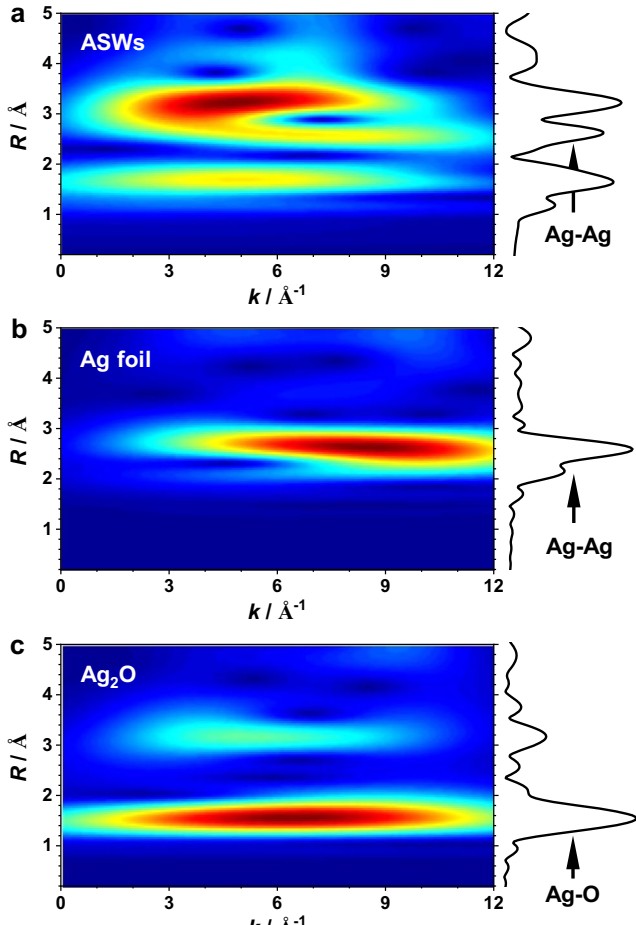

**Fig. 1 Local structures of Ag atoms in ASWs.** Wavelet transform plots and $k^3$-weighted EXAFS spectra in R-space of Ag (**a**), ASWs (**b**), and Ag₂O (**c**). Color scale: dark blue to red refers to low intensity to high intensity.

neighbor shells are attributed to Ag–O (~2.48 Å) and Ag–Ag (~2.87 Å) bonds with coordination numbers (CNs) of 4 and 2, respectively (Supplementary Fig. 4 and Table 3). These findings agree with the data of the SXRD refinements above. The average Ag–Ag bond length is close to that of 2.89 Å in bulk Ag (Supplementary Fig. 5 and Table 3)[12], implying that there is a strong Ag–Ag bonding in an ASW. The shortest Ag–O distance of 2.48 Å is significantly longer than the Ag–O bond length of 2.04 Å in Ag₂O (Supplementary Fig. 6 and Table 3), indicative of a weak interaction between ASWs and α-MnO₂.

The geometric structures of ASWs were directly imaged by high-resolution TEM (HRTEM). Figure 2a–c shows an HRTEM image, a simulated image[13,14], and a calculated diffraction pattern of ASWs viewed along an α-MnO₂[120] axis, respectively. In Fig. 2a, ASWs along the α-MnO₂[001] direction with the Ag–Ag bond length of ~2.87 Å are clearly visible, approaching the typical Ag–Ag bond length in bulk Ag. Similarly, parallel wires of bright atomic columns are observed along the α-MnO₂[111] direction in the high angle annular dark field scanning transmission electron microscopy (HAADF-STEM) image (Fig. 2d), particularly shown in the images after Fourier filtering (Fig. 2d inset and Fig. 2e)[14]. As the scattering cross section in HAADF-STEM image is approximately proportional to Z^1.8 (where Z is the atomic number)[15], Ag atoms appear brighter than Mn and O atoms. The bright linear atomic columns in the HAADF-STEM image thus

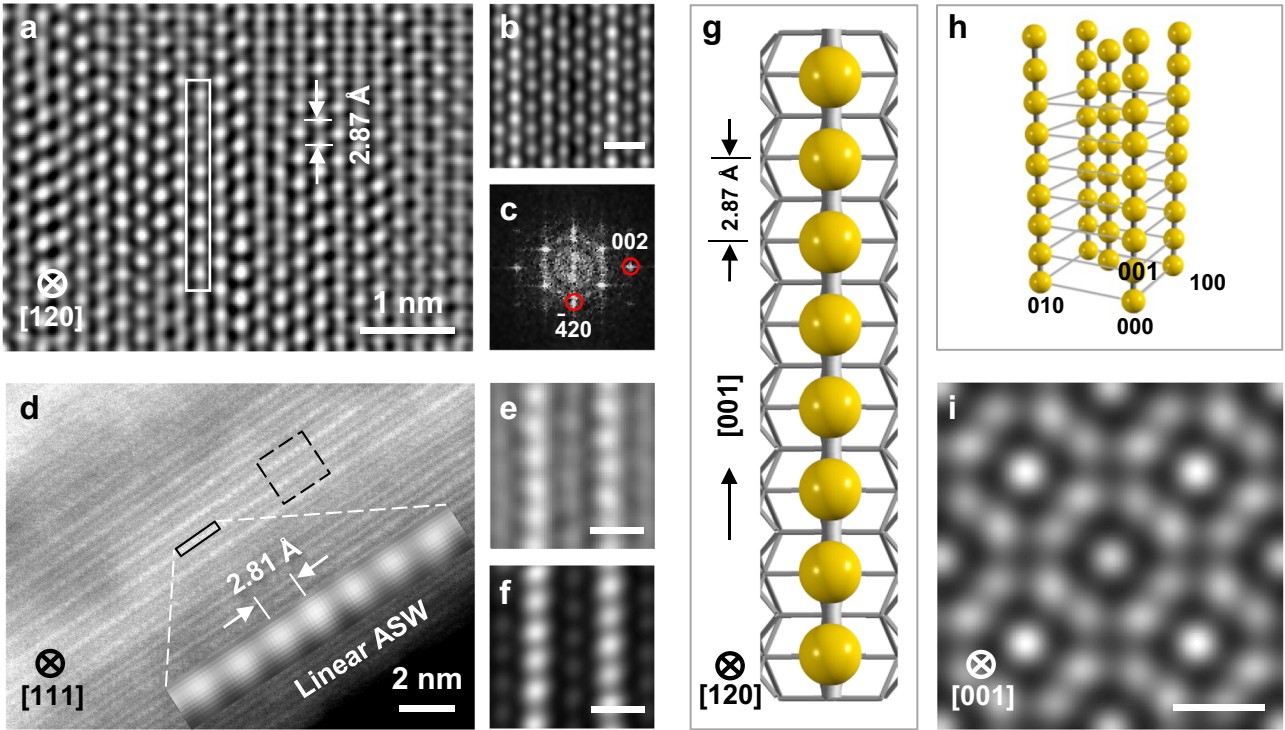

**Fig. 2 Electron microscopy imaging and structure models of ASWs. a**, **b** Filtered and simulated HRTEM images viewed from a α-MnO$_2$[120] axis. **c** Calculated diffraction pattern from **a**. **d** HAADF-STEM image, viewed from the α-MnO$_2$[111] direction (the inset shows an enlarged linear Ag wire after Fourier filtering). **e**, **f** Filtered and simulated images enlarged from the black dashed rectangle in **d**. **g** Structural model of one typical ASW together with the α-MnO$_2$ framework. **h** 3D model of the array with five ASWs. **i** Simulated image of ASWs in the α-MnO$_2$ tunnels viewed from the [001] direction showing the inter-wire distance. Scale bars in **b**, **e**, **f**, and **i**, 0.5 nm.

are ASWs, as shown by a corresponding simulated image (Fig. 2f). An observed distance of ~2.81 Å for a typical ASW is attributed to the projected distance of the Ag–Ag bond length of ~2.87 Å on the α-MnO$_2$(111) plane. These observations are in good accordance with the results of SXRD pattern and EXAFS spectra.

A structural model of the linear atomic wires was constructed on the basis of the above results (Fig. 2g). According to the experimental design, we filled all tunnels of the α-MnO$_2$ nanorods with Ag atoms to form the linear atomic Ag wires (Supplementary Fig. 2). As a result, the average length of ASWs is estimated to be ~0.5 μm with ~4% of ASWs longer than 1 μm (Supplementary Fig. 2). The average width of the α-MnO$_2$ nanorods is ~12 nm, and more than 280 parallel ASWs were assembled in the tunnels of each α-MnO$_2$ nanorod to form an atomic wire array (Fig. 2h). The closest-neighbor inter-wire spacing of this array is 6.95 Å (Fig. 2i). Hence, the suitable 3D sizes of this array enable ASWs as one electrical device to be connected into a circuit.

**Assembly process of the atomic wire array**. Next we present the assembly process of ASWs. We firstly prepared α-MnO$_2$ nanorods of up to 1 μm in length and over 10 nm in width, which have square 1D parallel tunnels extending along the [001] growth direction[16]. Ag/MnO$_2$ was then synthesized by depositing truncated octahedral Ag NPs onto the α-MnO$_2$ surfaces (inset of Fig. 3a, Supplementary Fig. 7). Normally, supported metal NPs are prone to sintering and forming larger particles through thermal activation[17,18] via Ostwald ripening and/or coalescence[19], depending on the particle size[20]. In our work, however, shrinking of Ag NPs on the α-MnO$_2$ surfaces was clearly observed by using temperature-programmed SXRD technique, in situ TEM, and EXAFS spectroscopy.

Figure 3a shows the temperature-programmed SXRD patterns of Ag/MnO$_2$. The intensity of the SXRD pattern of Ag NPs becomes weaker and ultimately disappears as the temperature increases, indicating that Ag NPs gradually shrink and finally reach a highly dispersive state. In particular, the Ag(111) reflection fades as the annealing temperature increases. At 383 K, Ag atoms become sufficiently active, detaching from Ag NPs and diffusing along the α-MnO$_2$ surfaces (Supplementary Fig. 8), similar to what has been observed for Ag atoms diffusing across Pt-Ag step boundaries[6]. As the temperature further increases to ~493 K, a clear anomaly in the first derivative SXRD patterns is observed (Supplementary Fig. 8), implying that the Ag atoms start to be assembled into the α-MnO$_2$ tunnels.

Recent work shows that the reactive environment plays a key role in the redispersion of noble metal NPs[21]. In order to understand the mechanism behind the novel process and rationalize the influence of atmospheres, we use an advanced in situ environmental TEM tool to record the structural evolution of MnO$_2$ supported Ag NPs in situ and with high spatial resolutions (Supplementary Figs. 9–10). Clearly, the annealing under the inert environment cannot trigger the re-dispersion of Ag NPs, but causes the serious aggregation. In contrast, the re-dispersion of Ag NPs took place during the oxygen annealing, leading to the formation of ASWs. This result keeps a good agreement with the temperature-programmed SXRD data. The in situ environmental TEM results evidence that the interaction between Ag NPs and O$_2$ is the driving force to enhance the mobility of Ag atoms[22], which move from the NPs to the support and are finally hosted in the matrix[23–25]. As showed in Supplementary Fig. 11 and corresponding Supplementary Movie, a Ag NP collapsed and quickly disappeared in 5 min at 270 °C in the presence of O$_2$. During the redispersion process, a stronger adherence of Ag NP to α-MnO$_2$ was observed, leading to the

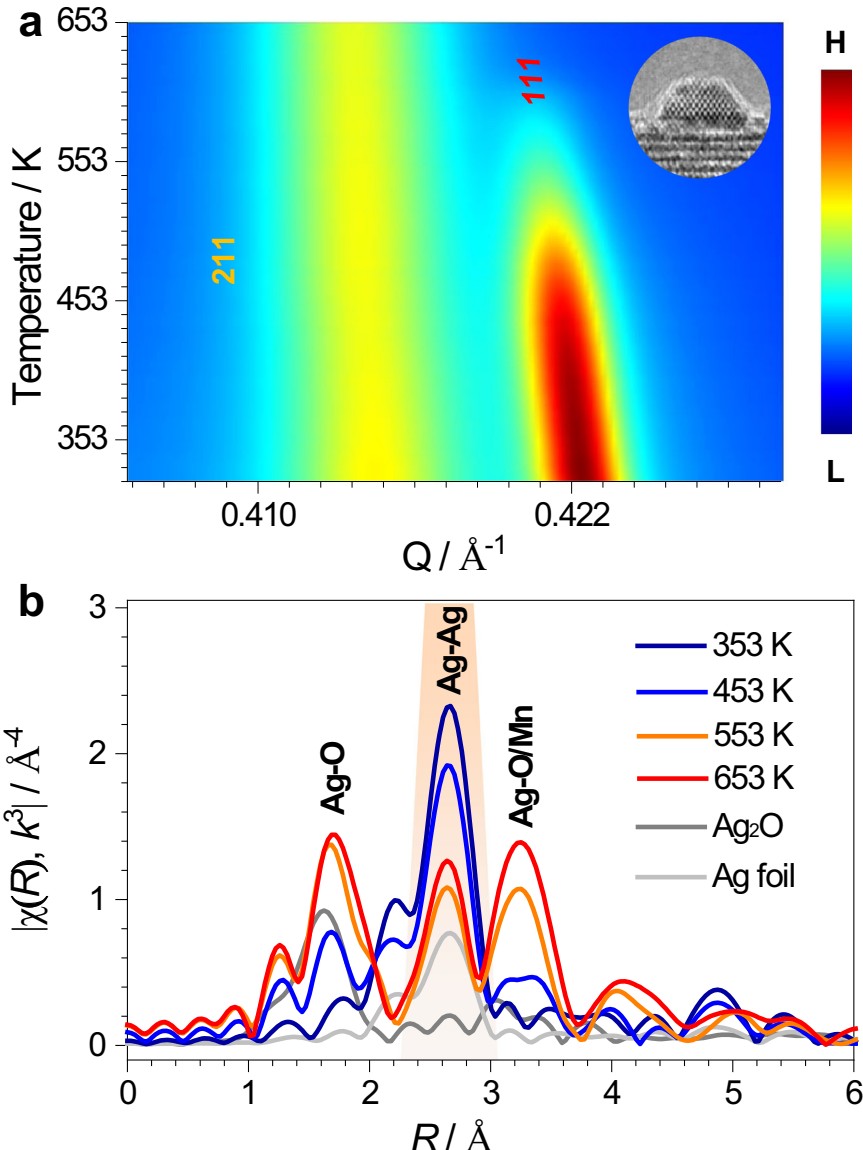

**Fig. 3 Assembly process of ASWs. a** Contour map of temperature-programmed SXRD patterns of Ag/MnO$_2$ as a function of momentum transfer (Q) from 323 to 653 K, showing the diffraction intensities of the Ag(111) and the α-MnO$_2$(211) reflections. Inset: HRTEM image of Ag/MnO$_2$ with a Ag NP of 2 nm in height and 4 nm in width. H and L in the scale bar refer to High and Low in intensity, respectively. **b** Room-temperature ex situ Ag K-edge χ(R) k$^3$-weighted FT EXAFS spectra of Ag/MnO$_2$ after being annealed at higher temperatures, and two references of Ag$_2$O and Ag foil. The amplitudes of Ag$_2$O, Ag foil, and Ag/MnO$_2$ spectra after being annealed at 353 K and 453 K were multiplied by 1/4, 1/18, 1/4, and 2/3, respectively.

collapse of the Ag NP. It looks like that the process is surface-mediated, in which atomic species after being emitted from a metal NP diffuse on the surface of the support until being trapped by a strong metal-support interaction[23–25] (Supplementary discussion).

The local structures of the Ag atoms during the thermal diffusion process were determined by using in situ EXAFS spectroscopy (Supplementary Fig. 12). To eliminate the effects of non-symmetric and inhomogeneous distribution of the instantaneous bond length owing to the thermal disorder[26], the corresponding ex situ EXAFS spectra were measured after the samples were annealed at higher temperatures and then cooled down to room temperature (Fig. 3b). As the annealing temperature increases, the Fourier transform (FT) amplitude due to the first Ag–Ag shell decreases, while the FT amplitudes due to two Ag–O shells increase. Specifically, after annealing at 653 K, the Ag atoms have been assembled into the tunnels to

form ASW arrays. Although the geometric size of the α-MnO$_2$ tunnel is 4.7 Å × 4.7 Å, the inner effective diameter is so small that it only allows single Ag atoms to diffuse along the tunnel at one time[27,28]. Thus, the atomic wire arrays are formed via the atom-by-atom assembly process.

**Electronic and electrical properties of the atomic wire array.** Finally, we explored the electronic structures and conducting properties of ASWs. Figure 4a shows the X-ray absorption near edge structure (XANES) spectra of ASWs, metallic Ag powder and Ag$_2$SO$_4$ at the Ag $L_3$ edge. Metallic Ag powder and Ag$_2$SO$_4$, respectively with the Ag$^0$($4d^{10}5s^1$) and Ag$^+$($4d^{10}5s^0$) electronic configuration are chosen as references in order to more precisely determine the electronic structures of ASWs[15]. No distinct peak at ~3354 eV appears in the XANES spectrum of metallic Ag because of the absence of the unoccupied 4d orbitals ($4d^{10}$). A

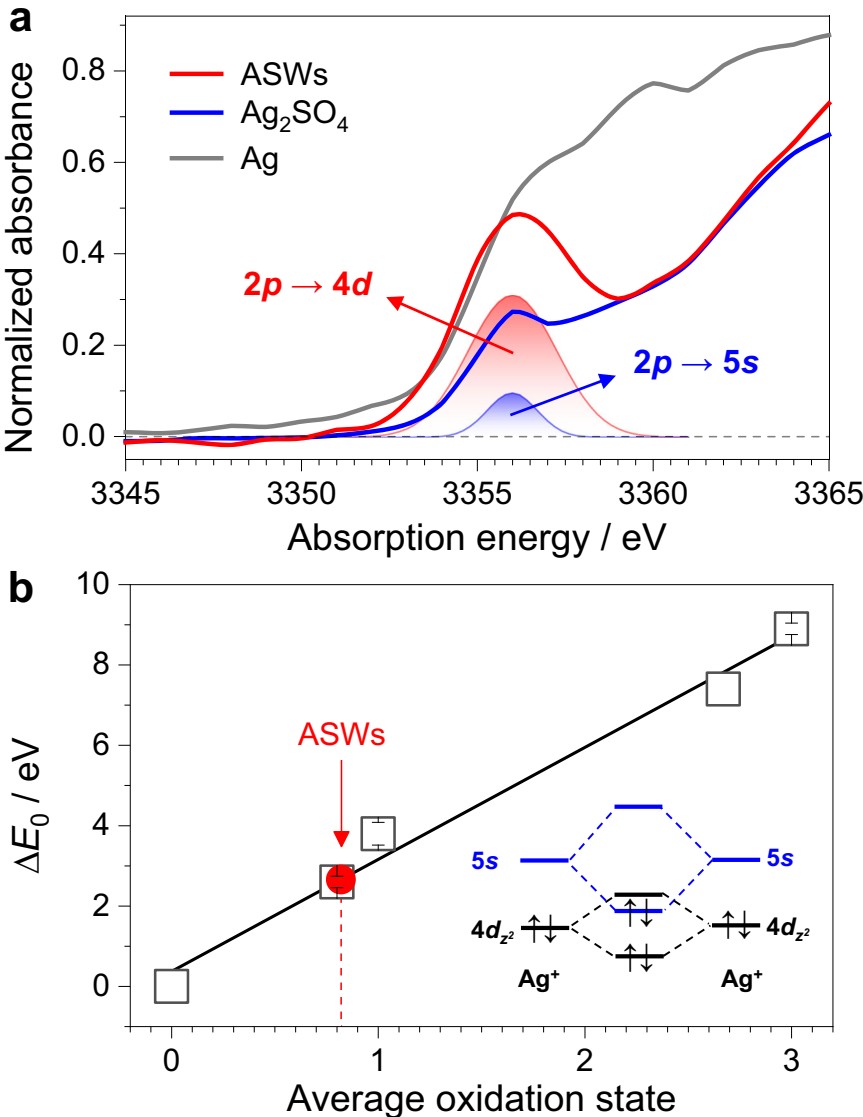

**Fig. 4 Electronic structures of ASWs. a** XANES spectra of ASWs and two references at the Ag $L_3$-edge. Red and blue shades represent the edge peaks of ASWs and $Ag_2SO_4$, respectively. **b** The change of Ag $L_1$-edge energy ($\Delta E_0$) versus average oxidation state. The Ag metal edge energy is selected as the reference energy, i.e., $\Delta E_0 = 0$ eV. Data of binary silver oxides containing Ag in formal oxidation states of +0.8, +1, +2.67, and +3 were obtained from reference [31]. Inset: Schematic illustration of the $4d$–$5s$ hybridization or overlapping among Ag atoms of ASWs leading to the $5s$ bonding orbitals shifting down below the $4d$ anti-bonding orbitals[15].

weak peak at 3354.8 eV was observed in the spectrum of $Ag_2SO_4$, which should be assigned to the $2p_{3/2} \rightarrow 5s$ transition and enhanced by the $4d$–$5s$ hybridization according to the fitting analysis (Fig. 4a)[15]. At the same absorption energy of 3354.8 eV, a strong peak appearing in the XANES spectrum of ASWs was mainly attributed to dipole-allowed $2p_{3/2} \rightarrow 4d$ transitions, suggesting the presence of the unoccupied Ag $4d$ orbitals[15]. The unoccupied orbitals are characteristics of anti-bonding orbitals, thus reinforcing Ag–Ag bonds[29,30]. Therefore, the reinforced Ag–Ag bonds together with the scaffolding function of α-$MnO_2$ via the weak Ag–O interactions enable ASWs to be very stable in air.

The electronic states of ASWs were further studied by using the XANES spectroscopy at the Ag $L_1$ edge (Supplementary Fig. 13), which can precisely determine the oxidation states[31]. Figure 4b shows the differential edge energies ($\Delta E_0$) with respect to that of metallic Ag as a function of average oxidation states. Thus, the average oxidation state of ASWs was determined to be close

to +1 (more precisely +0.82) according to the calibrated curve. On the basis of the results in Fig. 4a, the strong intensity of the edge peak reflects the hole in the $4d$ orbitals. By combining the results of XANES spectra at the Ag $L_3$ and $L_1$ edges, we deduced that ASWs might have a $Ag^{+0.8}$ ($4d^{9.25}5s^1$) electronic configuration. This evidence implies the presence of the $4d$–$5s$ hybridization, allowing the energy of the $5s$ bonding orbitals to be lower than that of the $4d$ anti-bonding orbitals. It is also not difficult to interpret this configuration by using a molecular orbital theory[15,31]. As calculated by Behrens[31], if the filled Ag $4d_{z^2}$ and empty $5s$ orbitals have the same symmetry and are hybridized, the $4d$ vacancy concomitant with the $5s$ occupied orbital is created (inset of Fig. 4b). As for ASWs, we set the wire direction as the $Z$ axis, and thus $4d_{z^2}$ and $5s$ orbitals have the same symmetry, resulting in the $4d_{z^2}$ vacancy. In principle, the $4d$ and $5s$ electrons are often characteristics of localized and delocalized features, respectively[32]. As a consequence, the depletion of the occupied states of the $4d_{z^2}$ anti-bonding orbitals strengthens the Ag–Ag

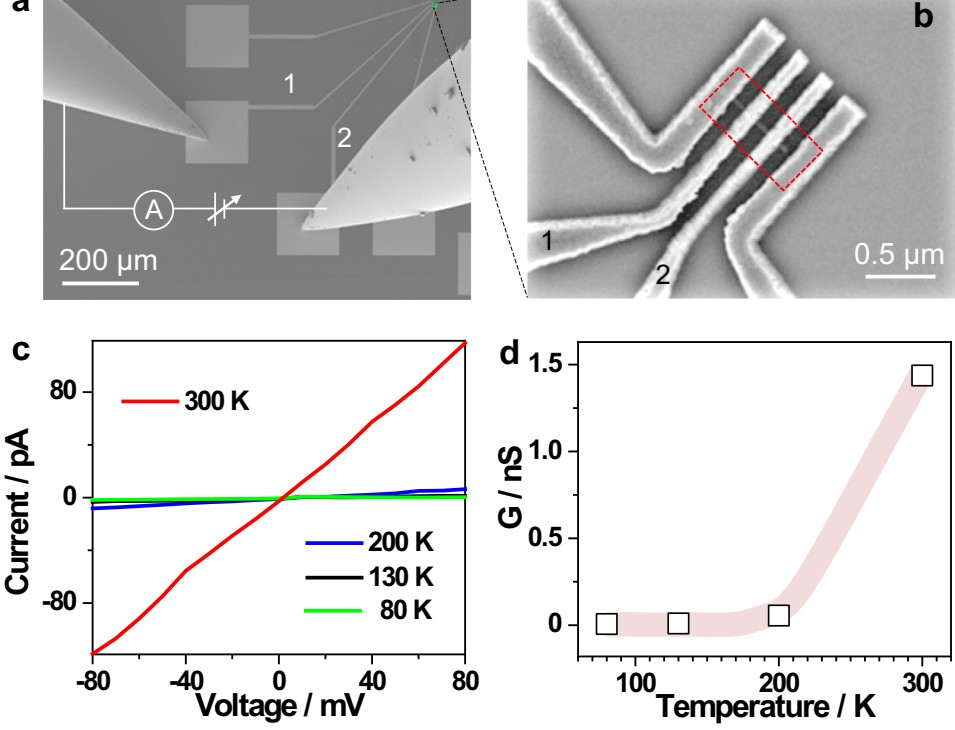

**Fig. 5 Electrical properties of ASWs. a** Circuit setup for the I–V measurements of ASWs. **b** An ASW array bridged between the nanoelectrodes (labeled as 1 and 2, highlighted in a red dashed rectangle), which was enlarged from the green rectangle in **a**. **c** The I–V curves of the ASW at different temperatures. **d** The corresponding conductance (G) of the array calculated from the slopes of the I–V curves (**c**).

bonds along the wire direction, greatly enhancing the stability of ASWs, and owing to the orbital spherical symmetry, the presence of the 5s electrons with the delocalized feature endows ASWs with the conducting property.

We measured the I-V curves of metallic ASWs in a temperature range of 80–300 K by bridging one typical atomic wire array between two nanoelectrodes to make an electrical circuit device (Fig. 5a, b). Figure 5c,d show the I–V curves of ASWs at different temperatures and the corresponding conductance as a function of temperature, respectively. ASWs exhibit thermally sensitive conductive properties of a temperature-dependent insulator-to-metal transition. Below 200 K, the current through ASWs is close to zero in the applied voltage range, and thus the conductance of ASWs approaches zero (Fig. 5d), implying an insulating behavior, i.e., the device is switched off. At 300 K, the linear I–V relation of ASWs demonstrates the Ohmic conductance (Fig. 5c) with a value of 1.5 nS (Fig. 5d) calculated from the slope of the red I–V curve in Fig. 5c, and, thus, the device is switched on. The I–V curve measured at 300 K confirms that electrons can be transferred along ASWs. Therefore, owing to the insulator-to-metal transition feature, ASWs can switch ON and OFF a circuit device, and this electrical feature allows the atomic wire array to be used as new thermal electrical devices.

## Discussion
To understand possible mechanisms behind the atomic assembling, we carried out the O 1s, Mn 2p X-ray photoelectron spectra (XPS) of the atomic wire array. Supplementary Figure 14 depicts O 1s XPS of Ag/MnO₂ and ASWs in the α-MnO₂ tunnels together with their difference, and only a little discrepancy in their O 1s XPS is discernible. Subtly, after Ag NPs were transferred into ASWs in the α-MnO₂ tunnels, the intensity of the O 1s XPS of MnO₂ slightly decreases in 532–534 eV, and increases at 530.5 eV,

suggesting that the electron density of the O atoms slightly increases after the formation of the atomic wire array in the α-MnO₂ tunnels. The oxidation states of Mn are preserved after the formation of ASWs (Supplementary Fig. 15). This indicates that the interaction mainly occurs between O atoms of α-MnO₂ and Ag of ASWs with a small amount of the Ag 4d electrons transferring to the O atoms of α-MnO₂, resulting in the partially unoccupied Ag 4d states, consistent with the results obtained from the XANES spectra above. The partial depletion of the electronic states is found at the top of the 4d orbitals with anti-bonding character[33], implying an increased strength of the Ag–Ag bonds[34]. As a result, α-MnO₂ has three main functions: (i) as a scaffold to protect ASWs from coalescence, (ii) as a special electron acceptor to only deplete the occupied states of Ag 4d anti-orbitals to strengthen the Ag–Ag bonds and keep the Ag 5s electrons with the delocalized states, and (iii) as an ideal container to assemble ASWs into the coherently oriented array with the suitable 3D sizes, which enables the array to be connected into thermal electrical devices. In fact, α-MnO₂ is not unique in the family of the porous materials, and we anticipate that other metal atomic wires or arrays can also be created by choosing suitable porous materials and optimizing host-guest interactions.

## Methods
**Sample synthesis.** All the chemicals are of analytical grade and used as received.

**Synthesis of ASWs.** α-MnO₂ was prepared by a hydrothermal route with an aqueous solution of MnSO₄ (0.20 mol L⁻¹), (NH₄)₂S₂O₈ (0.20 mol L⁻¹), and (NH₄)₂SO₄ (1.00 mol L⁻¹) at 393 K for 12 h. The obtained sample was filtered, washed with deionized water, dried at 383 K for 24 h, and calcined at 673 K for 4 h. The sample is expressed as AₓMn₈O₁₆ (x ≤ 2)[28], where A denotes the tunnel sites of α-MnO₂. AgNO₃ (0.733 g) was dissolved in de-ionized water to get a solution (60 mL) at room temperature, to which an aqueous ammonia solution (25 wt.%) was slowly added under stirring until the solution became transparent. Then, both the transparent solution and an H₂O₂ solution (30 wt.%, 30 mL) were simultaneously added to another suspension (80 mL) containing α-MnO₂ (1.760 g) under stirring

at 298 K for 1 h. The final suspension was filtered, washed with deionized water, and then dried in 353 K for 24 h to get Ag NPs supported on α-$MnO_2$ surfaces. Ag/$MnO_2$ was annealed at 653 K in air for 4 h to obtain ASWs inside α-$MnO_2$ tunnels.

**Transmission electron microscopy (TEM) images**. TEM, high resolution TEM (HRTEM) images, and energy-dispersive X-ray (EDX) microanalyses (point/line-scan analyses) were carried out with a JEOL JEM-2100F field-emission gun transmission electron microscope operated at an accelerating voltage of 200 kV and equipped with an ultra-high resolution pole-piece that provides a point-resolution better than 0.19 nm. It was also equipped with a STEM control unit (Gatan), EDX detector (SDD 80 $mm^2$), CCD camera (14-bit Gatan Orius SC600), bright-field (BF), and HAADF detectors (JEOL). Some microscope parameters used were as follows: defocus: −43.4 nm, Cs (spherical aberration): 0.5 mm, Cc (chromatic aberration): 1.0 mm. Fine powders of the materials were dispersed in ethanol, sonicated, and sprayed on a carbon coated copper grid, and then allowed to air-dry. Gatan SOLARUS 950 was used before observation. TEM simulations were conducted using the QSTEM v2.22 software based on multi-slice algorithm. A plane wave incident on the sample is modified by the projected potential of an atomic model partitioned into different slices. The modified complex wave is propagated through each slice of the sample.

**In situ TEM image**. We used an in situ gas holder (Climate S3, DENSsolutions) and a dedicated field-emission S/TEM (FEI Talos F20X) with an accelerating voltage of 200 kV to conduct the in-situ annealing experiments under different gaseous environments. The as-prepared Ag NPs supported on α-$MnO_2$ (Ag/$MnO_2$) was used as a precursor. The sample was directly dispersed in a MEMS chip for the in situ TEM experiment. After the holder equipped with the chip was inserted into TEM, different gases, $N_2$ or $O_2$, was introduced into the holder and the inner pressure was kept at 1 atm pressure. The temperature was gradually increased from room temperature to desired temperatures. Once the temperature reached at setting points, HAADF-STEM image of the sample was recorded. In order to reduce beam damage, electron beam was completely closed during the annealing treatment and the dose rate was minimized. Clearly, the imaging conditions allow for no damages to both Ag NPs or α-$MnO_2$. We also recorded the dynamic redispersion process of the supported Ag nanoparticles at 270 °C under atmospheric oxygen to visualize the deconstruction of the selected nanoparticles.

**Synchrotron X-ray diffraction (SXRD) patterns**. The room temperature SXRD patterns were recorded at BL14B of the Shanghai Synchrotron Radiation Facility (SSRF) at a wavelength of 0.6883 Å. A Mar345 image plate detector was employed for the data collection and the data were further integrated using the fit2d code[35]. The beam was monochromatized using a Si(111) crystal and a Rh/Si mirror was used for the beam focusing to a size of ~0.5 × 0.5 $mm^2$. Rietveld refinements of the diffraction data were performed with the FULLPROF software package on the basis of the space group $I4/m$. The temperature-programmed SXRD patterns were collected at BL14B of SSRF at a wavelength of 1.2398 Å. A typical amount of the sample (~1.5 mg) was loaded in a quartz capillary tube with a diameter of ~1 mm, and heating was carried out using a temperature-programmed procedure at a ramp of 3 K min$^{-1}$. Each SXRD pattern was collected at 3-min intervals and analyzed by using a CMPR software.

**X-ray absorption spectra**. X-ray absorption spectra at the Ag $K$-edge were measured at BL14W of the SSRF with an electron beam energy of 3.5 GeV and a ring current of 200–300 mA. The data were collected with a fixed exit monochromator using two flat Si(311) crystals. Harmonics were rejected by using a grazing incidence mirror. The XANES spectra at the Ag $L_{1,3}$-edge were acquired at BL4B7A of the Beijing Synchrotron Radiation Facility (BSRF) with an electron beam energy of 2.2 GeV and a ring current of 300–450 mA. The energy step for XANES measurement was set to be 0.2 eV. The extended X-ray absorption fine structure (EXAFS) spectra were collected in a transmission mode using ion chambers filled with $N_2$. The raw data were analyzed by using the IFEFFIT 1.2.11 software package.

**X-ray photoelectron spectra (XPS)**. XPS were recorded using Kratos Axis Ultra-DLD system with a charge neutralizer and a 150 W Al (Mono) X-ray gun (1486.6 eV) equipped with a delay-line detector (DLD). The spectra were acquired at a normal emission with a passing energy of 40 eV. The spectra were calibrated according to the C 1$s$ peak at 284.6 eV.

**Electrical measurements by in situ scanning electronic microscopy (SEM) technique**. Samples were dispersed in ethanol and dropped on a marked Si substrate with 300 nm $SiO_2$ layer. The position of the suitable individual wire array was determined by SEM observations (JSM-6700F). The Cr/Au (10 nm/100 nm) electrodes were patterned onto the top of the atomic wire array using lithography and electron-beam deposition followed by a lift-off process. Then, the wafer was transferred into a SEM with piezoelectric micromanipulators (Kleindiek). Finally, temperature-dependent electrical measurements were carried out by using a semiconductor characterization system (Keithley 4200) and a home-made cooling stage.

## Data availability

All the data that support the findings of this study are available from the corresponding author upon reasonable request. Source data are provided with this paper.

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

## Acknowledgements

We sincerely thank Prof. John Miao and Dr. Z. Amghouz for the help in the HAADF-STEM and EDX measurement, respectively. This work is supported by NSFC (21777030 and 21976037), the National Engineering Laboratory for Flue Gas Pollution Control Technology and Equipment (NEL-KF-201903), and the National Engineering Laboratory for Mobile Source Emission Control Technology (NELMS2018B02). D.T. acknowledges the support by JSPS Kakenhi (Grant JP20K05281, JP25820336) and World Premier International (WPI) Center for Materials Nanoarchitectonics (MANA) of the National Institute for Materials Science (NIMS), Tsukuba, Japan. D.G. is grateful to the Australian Research Council (ARC) for granting a Laureate Fellowship FL160100089. X.L. acknowledges the support by NSFC (21872163, 22072090, 21991153, 21991150). Both in situ SXRD patterns and Ag K–edge X-ray absorption spectra were conducted at the Shanghai Synchrotron Radiation Facility (SSRF), Shanghai, China. The Ag $L_{1,3}$-edge X-ray absorption spectrum measurements were conducted at the Beijing Synchrotron Radiation Facility (BSRF), Beijing, China.

## Author contributions

X.T. directed the project. Y.C. and Z.H. prepared the catalysts, conducted the experiments, and analyzed the data. D.T., J.C., T.S., Y.B. and D.G. conducted temperature-dependent electrical measurements. X.L. and X.W. conducted the in-situ TEM experiments. W.Q., J.C., D.X. and X.H. assisted with catalyst preparation and characterization. X.T., Y.C. and Z.H. wrote the manuscript. All authors commented on the manuscript.

## Competing interests

The authors declare no competing interests.
