## [Peer Review File · Nature Communications]

Reviewers' Comments:

Reviewer #1:

Remarks to the Author:

Though the utility of stable 1D single crystalline metal nanowires could be cardinal especially in the field of nanoelectronics, its experimental realization is often hindered due to high degree of instability arising from coordinative unsaturation at room temperature. In the present work, the authors have moderately demonstrated some plausible way to stabilize atomic silver wires (ASWs) with the aid of tunnel-structured α -MnO₂ nanorods. The novelty of this work lies apparently in the successful synthesis of stable atomic wires of appreciable length under ambient conditions as the manuscript focuses on exploiting the scaffolding function of α -MnO₂ nanorods to create stable ASWs, with signatures of temperature-driven metal-to-insulator transition, based on familiar self-assembly techniques.

The manuscript is well written and clearly organized with discussions related to all the process steps being largely unambiguous. Further, the experiments are also well performed while the results appear to be sound and important for the community. Thus, I would recommend publication in Nature Communications once the issues listed below are well addressed.

1. In the Abstract, the authors should consider adding little more about their essential observations on the electronic structure and electrical properties of the as-synthesized ASWs. Besides, some discussion of Ag-incorporated α -MnO₂ nanorods related to their modified electronic structure and electrical properties is also needed.
2. The morphology of α -MnO₂ nanorods, before and after the incorporation of silver nanoparticles, should be discoursed in order to highlight the morphological changes and associated effects on the formation of ASWs.
3. Authors should include in the manuscript some quantitative analysis of the growth parameters for ASWs such as the competing driving forces, the rate of growth, etc. What about the optical properties of such ASWs?
4. Did the authors consider using scanning surface potential microscopy to map the electric potential distribution of the as-synthesized ASWs?
5. The manuscript should be more carefully copyedited to correct certain typographical errors. For example, the word "section" is misspelt as "scetion" in page number 3 under the "Results" section.

Reviewer #2:

Remarks to the Author:

The authors synthesized the arrays of single atomic silver wires inside the parallel tunnels of α -MnO₂, which can be fabricated into actual devices for current-voltage characterizations. The formation of silver wires has been evidenced by synchrotron X-ray diffraction (SXRD) and EXAFS, indicating that the Ag-Ag bonds are formed in one-dimension. The electronic structure of ASWs was investigated by XANES, showing the characteristic 2p- \rightarrow 4d transition of ASWs. The I-V measurement shows the MI transition at \sim 200K potentially due to Peierls-like distortion.

It seems that the overall experimental data strongly support the conclusion of the manuscript, and the previous studies have been properly referenced. Thus, I recommend the publication of the manuscript after considering the following issues:

1. The authors emphasized the superiority of the ASWs compared with the previous studies, but some comparison is not correct. For example, the Ag nanowires in Ref. 8 are stable without

solvent, but they are stable only inside the organic scaffolds.

2. Electron energy loss spectroscopy (EELS) in TEM must be the best method to prove the existence of 1D Ag-Ag bond arrays shown in Fig.2d in comparison with bulk Ag or silver oxides particularly around the plasmon energies.

3. If possible, the comparison between I-Vg characteristics below and above Tc in Fig. 5c would be interesting potentially because of the band structure change induced by the lattice distortion.

Reviewer #3:

Remarks to the Author:

This manuscript reports the preparation and studies of atomic silver wires assemblies.

The presented experimental data and discussion are not sufficiently convincing and it is not clear how exactly the nanowires have been formed, and relevant mechanisms of the wire formation are not well elaborated. It is still hard to understand how the atomic Ag wires can be produced by a simple thermal diffusion process here. For example, it should be quite strong interaction (not weak Ag-O interactions as it is stated in the manuscript) of silver with oxygen atoms of MnO₂ during the thermal process resulting in the formation of silver oxide species. This would clearly affect the proposed atom-by-atom assembly process. In fact the average oxidation state of ASWs was determined to be close to +1 according to XANES spectroscopy. So the reasonable question is: "Are we dealing with silver oxide here?" It seems that the relevant understanding of chemistry at the nano-level is missing here. MnO₂ can not be an inert matrix in such conditions. In addition, ASWs also exhibited thermally sensitive conductive properties which are much more characteristic to metal oxides rather than to silver wires and comparison with atomic palladium wires stabilised with organic ligand are not really appropriate here as metal atoms have a completely different surrounding in these cases.

Thus, much more detailed studies and theoretical modelling are really necessary to understand what is really happening in these systems.

I. Reply to the reviewers

We thank the editor and the reviewers for carefully reviewing our manuscript. We have revised the manuscript carefully, according to the reviewers' comments. Below is a point-by-point response to the reviewers' comments.

Reviewer #1

Though the utility of stable 1D single crystalline metal nanowires could be cardinal especially in the field of nanoelectronics, its experimental realization is often hindered due to high degree of instability arising from coordinative unsaturation at room temperature. In the present work, the authors have moderately demonstrated some plausible way to stabilize atomic silver wires (ASWs) with the aid of tunnel-structured α -MnO₂ nanorods. The novelty of this work lies apparently in the successful synthesis of stable atomic wires of appreciable length under ambient conditions as the manuscript focuses on exploiting the scaffolding function of α -MnO₂ nanorods to create stable ASWs, with signatures of temperature-driven metal-to-insulator transition, based on familiar self-assembly techniques.

The manuscript is well written and clearly organized with discussions related to all the process steps being largely unambiguous. Further, the experiments are also well performed while the results appear to be sound and important for the community. Thus, I would recommend publication in Nature Communications once the issues listed below are well addressed.

Reply 1: Thank the reviewer for the encouraging comments.

Comment 1.1: *In the Abstract, the authors should consider adding little more about their essential observations on the electronic structure and electrical properties of the as-synthesized ASWs. Besides, some discussion of Ag-incorporated α -MnO₂ nanorods related to their modified electronic structure and electrical properties is also needed.*

Reply 1.1: Thank the reviewer for the good comments. We have revised the Abstract as the reviewer suggested: "Atomic metal wires, ideally with high stability in air at room temperature and a suitable length for convenient connection into nanocircuits,

are favorable for practical applications in electronic devices, but such atomic wires are extremely unstable and thus difficult to achieve. We fabricate stable atomic silver wires (ASWs) with the appreciable unoccupied states inside the parallel tunnels of α -MnO₂ nanorods. The unoccupied Ag 4d orbitals strengthen the Ag-Ag bonds, greatly enhancing the stability of ASWs, and the presence of the 5s electrons with the delocalized feature endows ASWs with the conducting property. These stable ASWs in the tunnels form a coherently oriented three-dimensional wire array of over 10 nm in width and up to 1 μ m in length, thus allowing us to connect it into a circuitry. Current-voltage characteristics of ASWs show a temperature-dependent insulator-to-metal transition, implying that the atomic wires could be used as thermal electrical devices.”

Comment 1.2: The morphology of α -MnO₂ nanorods, before and after the incorporation of silver nanoparticles, should be discoursed in order to highlight the morphological changes and associated effects on the formation of ASWs.

Reply 1.2: The synthesized α -MnO₂ usually has a nanorod-shaped morphology, and grows along a [001] direction (*Chem. Eur. J.* **2015**, *21*, 9619). As shown in Figure R1, the average width and length of α -MnO₂ nanorods before the incorporation of Ag nanoparticles (NPs) are 12 nm and 507 nm, respectively, which are almost same as those of α -MnO₂ nanorods after the incorporation of Ag (Figure R2). Thus, the morphology of α -MnO₂ nanorods hardly changed before and after the incorporation of Ag NPs. We have also added these data in the revised manuscript: “TEM data shows that the incorporation of Ag atoms hardly changed the morphology of α -MnO₂ nanorods (Supplementary Figs. 1 and 2). We further conducted a Rietveld refinement of room-temperature SXRD of ASWs inside the α -MnO₂ tunnels together with the pristine α -MnO₂ (Supplementary Fig. 3)¹².”

Figure R1. (a) TEM image of α -MnO₂ nanorods. (b) Width and length distributions of α -MnO₂ nanorods.

Figure R2. (a) TEM image of ASWs in α -MnO₂ tunnels. (b) Width and length distributions of ASWs. (c) Energy-dispersive X-ray spectroscopy and the corresponding elemental composition (inset table) of ASWs in α -MnO₂ tunnels (inset TEM image of an individual nanowire analyzed by EDX).

Comment 1.3: Authors should include in the manuscript some quantitative analysis of the growth parameters for ASWs such as the competing driving forces, the rate of growth, etc. What about the optical properties of such ASWs?

Reply 1.3: We appreciate the comments made by the reviewer. However, it should be extremely difficult to monitor the growth of single-atom chains inside the matrix due

to technical reasons. As showed in Figure R3 and corresponding video, a Ag NP collapsed and quickly disappeared in 5 minutes. It suggests that many tools, including scanning/transmission electron microscope (S/TEM), scanning tunneling microscope (STM) and X-ray absorption spectroscopy (XAS), cannot often track this dynamic process promptly. In order to understand the mechanism related with the re-dispersion process, we alternatively analyze the collapse of Ag NPs under the reaction conditions by using *in-situ* XRD and *in-situ* environmental TEM techniques. Therefore, the decrease in volume of a single Ag NP as a function of calcination time under the reaction conditions was displayed in Figure R4a. Clearly, the shrink of the NP size could be distinguished into three distinctive stages: initially, the NP's volume decreased slowly with the particle size larger than about 13 nm; afterwards, the NP's volume dropped rapidly with the particle size ranging from 13 to 10 nm; when the particle was smaller than 10 nm, the shrink became slower until the whole particle disappeared. It suggests that the re-dispersion process might be modulated by multi factors, which played different roles at these stages. Presumably, when the particle size became smaller, the strong metal-support interaction (SMSI) may play a main role to hinder the immigration of Ag atoms out of the parent particle. A similar trend was also observed in the *in-situ* XRD analysis as shown in Figure R4b, which indicates the change in Ag₁₁₁ peak area as a function of calcination temperature.

Figure R3. A serial of *in-situ* TEM images of Ag/MnO₂ recorded at 270 °C in the presence of O₂ (see Supplementary Video 1). Clearly, during the re-dispersion process, a stronger adherence of the Ag NP to α -MnO₂ was observed, leading to the collapse of the Ag NP. It looks like that the process is surface-mediated, in which atomic species after being emitted from a metal NP diffuse on the surface of the support until being

trapped by a strong metal-support interaction (*Nat. Nanotechnol.* **2019**, *14*, 851; *Nat. Commun.* **2019**, *10*, 234; *Nat. Commun.* **2020**, *11*, 1263).

Figure R4. (a) The decrease in volume of the single Ag NP regarded as spherical as a function of calcination time (270 °C, atmospheric O₂ environment, also see Figure R3 and Supplementary video 1). (b) Integrated area of Ag₁₁₁ peak as a function of calcination temperature (Air, 3 °C min⁻¹), which was obtained from the *in-situ* XRD patterns of Supplementary Figure 8.

In order to understand the mechanism behind the novel process and rationalize the influence of atmospheres, we use an advanced *in-situ* environmental TEM tool to record the structural evolution of α MnO₂ supported Ag NPs and with high spatial resolutions (see Figures R5-6). Clearly, the annealing under the inert environment such as N₂ cannot trigger the re-dispersion of Ag NPs, but causes the serious aggregation. In contrast, the re-dispersion of Ag NPs took place during the oxygen annealing, leading to the formation of ASWs. This result keeps a good agreement with the *in-situ* XRD result. The *in-situ* environmental TEM results evidence that the interaction between Ag NPs and O₂ should be the driving force to enhance the mobility of Ag atoms (*Angew. Chem. Int. Ed.* **2017**, *56*, 13078), which moves the Ag atoms from the NPs to the support and these detached Ag atoms are finally hosted in the matrix (*Nat. Nanotechnol.* **2019**, *14*, 851; *Nat. Commun.* **2019**, *10*, 234; *Nat. Commun.* **2020**, *11*, 1263). However, the HRTEM image of the intermediates suggests that the process is surface-mediated, in which surface oxygen atoms also interact with

detached Ag atoms and modulate their transportation. (*Nat. Nanotechnol.* **2019**, *14*, 851; *Nat. Commun.* **2019**, *10*, 234; *Nat. Commun.* **2020**, *11*, 1263). Apparently, when the destruction rate of Ag NPs was not constant and changed in different circumstances related with particle sizes and the interaction between the support and NPs, the growth parameters for ASWs would change consequently. Therefore, we insert related discussion in the revised manuscript and Supplementary information, which will help readers to understand the complexity behind the formation of ASWs.

Figure R5. The *in-situ* HAADF-STEM images of Ag/MnO₂ recorded at different temperatures during the heating treatment under atmospheric N₂ environment. The *in-situ* data show that bright silver NPs aggregated into larger NPs with the increase of temperature.

Figure R6. The *in-situ* HAADF-STEM images of Ag/MnO₂ recorded at different temperatures during the heating treatment under atmospheric O₂ environment. Clearly, all Ag NPs disappeared above 250 °C in the presence of O₂. In order to examine if silver atoms diffused into the matrix or immigrated to other places, the temperature was increased to 750 °C to destroy the structure of α -MnO₂. It can be seen that the morphologies of α -MnO₂ nanorods changed greatly and the hosted silver atoms were released and aggregated into larger NPs again.

As suggested by the reviewer, the optical properties of α -MnO₂ and ASWs were analyzed by diffuse reflectance spectroscopy (DRS), and Figure R7 shows the corresponding results. Both samples show a wide absorption band ranging from 200 to 550 nm, which is similar to the result of the reference (*J. Phys. Chem. C* **2008**, *112*, 13134). We also estimated the optical band gap energy for both samples by Kubelka-Munk model ($\alpha hv = A(hv - E_g)^{1/n}$, where α is absorption coefficient; h is Plank's constant; v is light frequency and E_g is the band gap energy. Hence, the value of n is calculated to be approximately 2 in this case according to the reference (*J. Phys. Chem. C* **2008**, *112*, 13134); By plotting the $(\alpha hv)^{1/2}$ versus hv , the band gap energies for ASWs and α -MnO₂ can be derived as 1.58 and 1.49 eV, respectively.

Though these optical properties of α -MnO₂ and ASWs are interesting, we did not added them into the revised manuscript owing to the small difference between them.

Figure R7. (a) Diffusive reflectance absorption spectra of ASWs and α -MnO₂. (b) the $(\alpha hv)^{1/2}$ vs hv plot.

Comment 1.4: Did the authors consider using scanning surface potential microscopy to map the electric potential distribution of the as-synthesized ASWs?

Reply 1.4: Thank the reviewer for the suggestion. Scanning surface potential microscopy (SSPM) is an advanced tool to measure the local contact potential difference between a conducting atomic force microscopy tip and the sample, thereby mapping the work function or surface potential of the sample with high spatial resolutions. However, the demands for sample preparation are crucially high for scanning surface potential microscopy characterization (*Thin Solid Films* **2009**, 517, 5100). In particularly, the as-prepared ASWs were inside the parallel tunnels ($4.7 \text{ \AA} \times 4.7 \text{ \AA}$) of α -MnO₂ nanorods, which grow along [001] direction and expose {100} and {010} surfaces (Figure R8) (*Chem. Eur. J.* **2015**, 21, 9619). A distance between neighbor ASWs is only about 4.7 \AA . Thus, how to measure the electric potential distribution of embedded Ag atoms and their oxide circumstance at atomic scales brings great challenges to the current characterization techniques. Considering the technical difficulties and challenges, we will collaborate with other laboratories, which are more experienced in SSPM, to develop this technique and methodology in the future.

Figure R8. The model of ASWs inside the α -MnO₂ nanorods from different directions. Yellow, red, and purple balls represent Ag, O, and Mn, respectively.

Comment 1.5: The manuscript should be more carefully copyedited to correct certain typographical errors. For example, the word “section” is misspelt as “section” in page number 3 under the “Results” section.

Reply 1.5: Thank the reviewer for the careful review. We have revised the manuscript carefully and corrected the typographical errors.

Reviewer #2

The authors synthesized the arrays of single atomic silver wires inside the parallel tunnels of α -MnO₂, which can be fabricated into actual devices for current-voltage characterizations. The formation of silver wires has been evidenced by synchrotron X-ray diffraction (SXRD) and EXAFS, indicating that the Ag-Ag bonds are formed in one-dimension. The electronic structure of ASWs was investigated by XANES, showing the characteristic 2p→4d transition of ASWs. The I-V measurement shows the MI transition at ~200K potentially due to Peierls-like distortion.

It seems that the overall experimental data strongly support the conclusion of the manuscript, and the previous studies have been properly referenced. Thus, I recommend the publication of the manuscript after considering the following issues:

Reply 2: Thank the reviewer for the positive comments.

Comment 2.1: The authors emphasized the superiority of the ASWs compared with the previous studies, but some comparison is not correct. For example, the Ag

nanowires in Ref. 8 are stable without solvent, but they are stable only inside the organic scaffolds.

Reply 2.1: Thank the reviewer for the good comment. We have deleted this comparison in the revised manuscript.

Comment 2.2: *Electron energy loss spectroscopy (EELS) in TEM must be the best method to prove the existence of 1D Ag-Ag bond arrays shown in Fig.2d in comparison with bulk Ag or silver oxides particularly around the plasmon energies.*

Reply 2.2: Electron energy loss spectroscopy (EELS) in TEM is a good tool providing access to a wealth structural, chemical and physical information at the nanoscale. The low-loss EELS spectra particularly around the plasmon energies provide the information about the band structure and the dielectric properties of a material, from which we can also get the optical spectroscopy and valence-band X-ray photoelectron spectroscopy (*IOP Conf. Ser.: Mater. Sci. Eng.* **2016**, *109*, 012007; *Phys. Rev. B* **2000**, *62*, 11126). The core-loss EELS spectra provide similar information to that provided by XAS, including elemental quantification, bonding analysis, and local environment (*IOP Conf. Ser.: Mater. Sci. Eng.* **2016**, *109*, 012007; *Rep. Prog. Phys.* **2008**, *72*, 016502). However, considering the low signal-to-noise ratio of the elemental signal of EELS which is mainly due to the high uncharacteristic background below the ionization edges and the low ionization cross-sections for heavier elements and for elements occurring at low concentrations, XAS may be a better and easier way to measure the local environment of Ag atoms in our system. As shown in Figure 3b and Supplementary Figures 3-5, for the sample after annealing at 653 K, the interatomic distances in the two nearest-neighbour shells were attributed to Ag–Ag (~2.87 Å) and Ag–O (~2.48 Å) distances with CNs of 2 and 4, respectively. The average Ag–Ag distance is close to the Ag–Ag bond length of 2.89 Å in bulk Ag. Hence, the Ag atoms have been assembled into the tunnels to form ASW arrays.

Comment 2.3: *If possible, the comparison between I-Vg characteristics below and above Tc in Fig. 5c would be interesting potentially because of the band structure*

change induced by the lattice distortion.

Reply 2.3: Thank the reviewer for the good suggestion. Due to the pandemic, particularly, Japan's COVID-19 crisis worsens and one researcher in our collaborator's lab is infected with the virus, which leads to the shut-down of the lab. Hence, it is very regretful that we cannot finish this interesting experiment within the three-month revision period, but we think that it should be one of the important topics for further study when the lab reopens in the future.

Reviewer #3

Comment 3.1: *This manuscript reports the preparation and studies of atomic silver wires assemblies. The resented experimental data and discussion are not sufficiently convincing and it is not clear how exactly the nanowires have been formed, and relevant mechanisms of the wire formation are not well elaborated. It is still hard to understand how the atomic Ag wires can be produced by a simple thermal diffusion process here.*

Reply 3.1: Thank the reviewer for the good comments. Recent work shows that high-temperature synthesis is an effective method to produce thermally stable single atom catalysts (*Science* **2016**, 353, 150; *Nat. Nanotechnol.* **2018**, 13, 856; *Nat. Nanotechnol.* **2019**, 14, 851; *Nat. Commun.* **2019**, 10, 234; *Nat. Commun.* **2020**, 11, 1263), which suggests several mechanisms of metal dispersion. We noticed that the reactive environment plays a key role in the re-dispersion of noble metal NPs (*Nat. Catal.* **2019**, 2, 955). In order to understand the mechanism behind the novel process and rationalize the influence of atmospheres, we used an advanced *in-situ* environmental TEM tool to record the structural evolution of αMnO_2 supported Ag NPs and with high spatial resolutions (see Figures R5-6). Clearly, the annealing under the inert environment such as N_2 cannot trigger the re-dispersion of Ag NPs, but causes the serious aggregation. In contrast, the re-dispersion of Ag NPs took place during the oxygen (O_2) annealing, leading to the formation of ASWs. This result keeps a good agreement with the *in-situ* XRD result. The *in-situ* environmental TEM results evidence that the interaction between Ag NPs and O_2 is the driving force to

enhance the mobility of Ag atoms (*Angew. Chem. Int. Ed.* **2017**, *56*, 13078), which moves the Ag atoms from the NPs to the support and finally these mobile Ag atoms are hosted in the matrix (*Nat. Nanotechnol.* **2019**, *14*, 851; *Nat. Commun.* **2019**, *10*, 234; *Nat. Commun.* **2020**, *11*, 1263).

All the results above evidence the important role of O₂ and disclose the surface-mediated mechanism of Ag dispersion in our case and we also added these data into the revised manuscript.

Comment 3.2: *For example, it should be quite strong interaction (not weak Ag–O interactions as it is stated in the manuscript) of silver with oxygen atoms of MnO₂ during the thermal process resulting in the formation of silver oxide species. This would clearly affect the proposed atom-by-atom assembly process.*

Reply 3.2: We agreed with the reviewer that a quite strong interaction between Ag atoms and oxygen atoms of α -MnO₂ exists during the thermal diffusion process. According to the reviewer's suggestion, we revised the manuscript as: "The oxidation states of Mn are preserved after the formation of ASWs (Supplementary Fig. 15). This indicates that the interaction mainly occurs between O atoms of α -MnO₂ and Ag of ASWs with a small amount of the Ag 4d electrons transferring to the O atoms of α -MnO₂, resulting in a partially unoccupied Ag 4d states, consistent with the results obtained from the Ag 3d XPS and XANES spectra above. The partial depletion of the electronic states is found at the top of the 4d orbitals with anti-bonding character,³³ implying an increased strength of the Ag–Ag bonds."³⁴

Comment 3.3: *In fact, the average oxidation state of ASWs was determined to be close to +1 according to XANES spectroscopy. So the reasonable question is: "Are we dealing with silver oxide here?" It seems that the relevant understanding of chemistry at the nano-level is missing here.*

Reply 3.3: Although the average oxidation state of ASWs is close to +1, the geometric and electronic structures of ASWs are extremely different from those of Ag₂O. According to the EXAFS, XRD, and STEM data, we successfully constructed

the structural model of ASWs (Figure 2), i.e., single silver wires were inside the tunnels of the α -MnO₂ rod with the Ag-Ag bond length of ~ 2.87 Å and the Ag-O bond length of ~ 2.48 Å. The coordination numbers for Ag-Ag and Ag-O are 2 and 4, respectively. However, Ag₂O has the cubic cuprite crystal structure with a Ag-O bond length of ~ 2.04 Å and without any Ag-Ag bond (*J. Phys. Chem. A* **2010**, *114*, 4093). Furthermore, for ASWs, XANES spectra of ASWs at the Ag *L*₃-edge implies the depletion of the occupied states of the 4*d*_{z²} anti-bonding orbitals which strengthens the Ag-Ag bonds along the wire direction, greatly enhancing the stability of ASWs. Owing to the orbital spherical symmetry, the presence of the 5*s* electrons with the delocalized feature endows ASWs with the conducting property.

Comment 3.4: *MnO₂ cannot be an inert matrix in such conditions. In addition, ASWs also exhibited thermally sensitive conductive properties which are much more characteristic to metal oxides rather than to silver wires.*

Reply 3.4: We measured the current-voltage (*I-V*) characteristics of one individual α -MnO₂ rod at 300 K (Figure R9a). In Figure R9b, no electrical current is detected for the pure α -MnO₂ rods, indicating the insulating property of α -MnO₂. As a result, the conductive properties should originate from the silver atomic wires.

Figure R9. (a) TEM image of the electrical measurement configurations for a

α -MnO₂ rod. Nanometer Au and W tips were used as electrodes. (b) The corresponding *I-V* curve of the α -MnO₂ rod in (a).

Comment 3.5: *Comparison with atomic palladium wires stabilized with organic ligand are not really appropriate here as metal atoms have a completely different surrounding in these cases.*

Reply 3.5: Thank the reviewer for the good comment. We have deleted this comparison in the revised manuscript.

Comment 3.6: *Thus, much more detailed studies and theoretical modelling are really necessary to understand what is really happening in these systems.*

Reply 3.6: Recent work shows that high-temperature synthesis is an effective method to produce thermally stable single atom catalysts (*Science* **2016**, 353, 150; *Nat. Nanotechnol.* **2018**, 13, 856; *Nat. Nanotechnol.* **2019**, 14, 851; *Nat. Commun.* **2019**, 10, 234; *Nat. Commun.* **2020**, 11, 1263), which include two kinds of mechanism of metal dispersion. One is the gas-phase atom trapping mechanism, in which the volatile species are emitted from a metal NP and are trapped by a support that can bind the mobile species (*Science* **2016**, 353, 150). The other is the surface-mediated mechanism, in which atomic species after being emitted from a metal NP diffuse on the surface of the support until being trapped by a strong metal-support interaction (*Nat. Nanotechnol.* **2019**, 14, 851; *Nat. Commun.* **2019**, 10, 234; *Nat. Commun.* **2020**, 11, 1263). Considering that there are usually metal losses during the gas-phase atom trapping process (*Nat. Catal.* **2018**, 1, 781; *Nat. Nanotechnol.* **2018**, 13, 856) and that no detectable Ag loss was observed after the calcination at 653 K, it should be the surface-mediated mechanism of Ag dispersion in our case. Therefore, there would be two steps to synthesize ASWs from the supported Ag NPs: 1) single Ag atoms detach from Ag NPs and diffuse on the α -MnO₂ surfaces; 2) Ag atoms start to be assembled into the α -MnO₂ tunnels and diffuse in the tunnels.

1) Single Ag atoms detach from Ag NPs and diffuse on the α -MnO₂ surfaces.

Due to the low Hüttig temperature of Ag (*Nanotechnology* **2006**, 17, 3518) and

the liquid like behavior of Ag particles smaller than 10 nm (*Nat. Mater.* **2014**, *13*, 1007-1012), it is possible that Ag atoms become sufficiently active, detaching from Ag NPs at ~ 383 K (Supplementary Figure 8). Shrinking of Ag NPs on the α -MnO₂ surfaces was clearly observed by using temperature-programmed SXRD technique and *in-situ* TEM tool. This process has been discussed in the manuscript and Response 3.1 in detail. All the results above evidence the important role of O₂ and also disclose the surface-mediated mechanism of Ag dispersion in our case.

2) Ag atoms start to be assembled into the α -MnO₂ tunnels and diffuse in the tunnels.

Although it is hard to measure or record this process, according to the first derivative of the *in-situ* SXRD patterns, a clear anomaly is observed as the temperature increases up to ~ 493 K (Supplementary Figure 8), implying that the Ag atoms start to be assembled into the α -MnO₂ tunnels. We also calculated the diffusion barrier for the Ag atom diffusing from site to site in the tunnels. As shown in Figure R10, the barrier of such a diffusion inside the tunnels is so small (only ~ 0.31 eV) that it is relatively easy for Ag atoms to be assembled into ASWs in the α -MnO₂ tunnels.

Figure R10. Diffusion path of an Ag atom inside the α -MnO₂ tunnels. Inset: An atom model showing the diffusion path. Yellow, red, and purple balls represent Ag, O, and Mn, respectively.

Furthermore, we also revised the related descriptions and corrected the typographical errors all over the manuscript carefully.

Reviewers' Comments:

Reviewer #1:

Remarks to the Author:

The authors have addressed all the main points that I had pointed out in the review report. They have also improved the manuscript to a great extent by answering to the questions, comments and suggestions of all referees. In my opinion, it can be accepted now for publication.

Reviewer #2:

Remarks to the Author:

The authors have revised the manuscript properly according to the reviewers' comments. Thus, I recommend the publication of the manuscript in Nature Communications without further review.

Reviewer #3:

Remarks to the Author:

The authors have revised the manuscript appropriately and addressed all issues. Therefore the manuscript can be published in revised form.

Response to the reviewers

We thank the reviewers for carefully reviewing our manuscript. Below is a point-by-point response to the reviewers' comments.

Reviewer #1

The authors have addressed all the main points that I had pointed out in the review report. They have also improved the manuscript to a great extent by answering to the questions, comments and suggestions of all referees. In my opinion, it can be accepted now for publication.

Response 1: Thank you very much for your previous valuable comments, positive evaluation and publication recommendation on this work.

Reviewer #2

The authors have revised the manuscript properly according to the reviewers' comments. Thus, I recommend the publication of the manuscript in Nature Communications without further review.

Response 2: Thank you very much for your positive evaluation, all the previous valuable comments and publication recommendation on this work.

Reviewer #3

The authors have revised the manuscript appropriately and addressed all issues. Therefore the manuscript can be published in revised form.

Response 3: Thank you very much for your positive evaluation, all the previous valuable comments and publication recommendation on this work.